Unidirectional hybridization between American paddlefish Polyodon spathula (Walbaum, 1792) and sterlet Acipenser ruthenus (Linnaeus, 1758)

Káldy Jenő kaldy.jeno@uni-mate.hu 1
Fazekas Georgina 1 2
Kovács Balázs 3
Molnár Mariann 2 4
Lázár Bence 4 5
Pálinkás-Bodzsár Nóra 4
Ljubobratović Uroš 1
Fazekas Gyöngyvér 1
Kovács Gyula 1
Várkonyi Eszter 4
1 Research Centre for Fisheries and Aquaculture, Institute of Aquaculture and Environmental Safety, Hungarian University of Agriculture and Life Sciences , Szarvas , Békés , Hungary
2 PhD School of Animal Biotechnology and Animal Science, Hungarian University of Agriculture and Life Sciences , Gödöllő , Pest , Hungary
3 Department of Molecular Ecology, Institute of Aquaculture and Environmental Safety, Hungarian University of Agriculture and Life Sciences , Gödöllő , Pest , Hungary
4 National Centre for Biodiversity and Gene Conservation, Institute for Farm Animal Gene Conservation , Gödöllő , Pest , Hungary
5 Animal Biotechnology Department, Institute of Genetics and Biotechnology, Hungarian University of Agriculture and Life Sciences , Gödöllő , Pest , Hungary
Esteban María Ángeles
Electronic publication date: 2024 Jan 19
Publication date: 2024
Volume: 12
Electronic Location ID: e16717
Received 2023 Jun 23; Accepted 2023 Dec 4
Copyright: ©2024 Káldy et al.
Copyright year: 2024
Copyright holder: Káldy et al.
License: This is an open access article distributed under the terms of the Creative Commons Attribution License, which permits unrestricted use, distribution, reproduction and adaptation in any medium and for any purpose provided that it is properly attributed. For attribution, the original author(s), title, publication source (PeerJ) and either DOI or URL of the article must be cited.
License URL: https://creativecommons.org/licenses/by/4.0/

Keywords: Paddlefish, Sturgeon, Embryo development, Unidirectional hybridization, Unviable hybrid embryo

Funding: Research Centre for Fisheries and Aquaculture, Institute of Aquaculture and Environmental Safety, Hungarian University of Agriculture and Life Sciences, (Hungary) This work was funded by the Research Centre for Fisheries and Aquaculture, Institute of Aquaculture and Environmental Safety, Hungarian University of Agriculture and Life Sciences, (Hungary). The funders had no role in study design, data collection and analysis, decision to publish, or preparation of the manuscript.

==============================
Interspecific hybridizations among sturgeon species are feasible and often bidirectional. The American paddlefish (Polyodon spathula) from Family Polyodontidae and sturgeon species from Family Acipenseridae were reported capable of hybridization, but viable hybrids have been described only in crosses with the American paddlefish as paternal parents. In the reciprocal cross, the hybrids were not viable however embryos start to develop and reach late gastrula and early neurula stages. The goal of this study was to examine the hybridization between the sterlet sturgeon (Acipenser ruthenus) and the American paddlefish. Hybrid and purebred crosses were produced by artificial fertilization. Viable hybrid offspring were harvested (three month old) and verified in the families produced by female sterlet crossing with male American paddlefish. In the reciprocal hybrid crosses with female American paddlefish and male sterlet, the embryos development did not pass over 120 h post fertilization, indicating the unidirectional hybridization between American paddlefish and sterlet. Chromosome counting showed for the same ploidy level of viable hybrid and parent species. Analysis of three microsatellite markers confirmed the unidirectional hybridization between the American paddlefish and the sterlet species. Overall, the inferred genetic cause suggests that unidirectional hybridization between American paddlefish and sterlet may be the case not only for these two species but likely also between American paddlefish and other sturgeon species.

Introduction

Acipenserid fishes have a high rate of interspecific hybridization (Birstein, Hanner & De Salle, 1997). Family Acipenseridae, comprising the sturgeons, is the only vertebrate family in which all species can hybridize with each other in their natural overlapping habitats (Birstein & De Salle, 1998). However, hybrids between sturgeon from the family Acipenseridae with other Acipenseriformes family (Polyodontidae) are hardly known (Káldy et al., 2020). The diploid chromosome number of the ancient common ancestor of acipenseriform was approximately 60 (Birstein, Hanner & De Salle, 1997; Ludwig et al., 2001). During the evolution of acipenseriform, several polyploidization events occurred and resulted in three ploidy levels in sturgeons and paddlefishes: (1) 120 chromosomes; (2) 250–270 chromosomes and (3) 370 chromosomes (Vasil’v et al., 2014). Depending on the maximum ploidy levels, during the evolution of each species, can be distinguished the evolutionary and the current functional scale of the chromosome number for all acipenseriform species (Rajkov, Shao & Berreb, 2014). The evolutionarily tetraploids with approximately 120 chromosomes function as diploids and the evolutionarily octoploids with approximately 240 chromosomes function as tetraploids and the evolutionarily dodecaploids with approximately 360 chromosomes function as hexaploids (Havelka et al., 2011). In this article, the functional ploidy levels were applied as described above).

Successful hybridization between taxonomically distant species is rare due to genetic incompatibility of different chromosome numbers (Arnold, 1997; Parker, Philipp & Whitt, 1985a; Parker, Philipp & Whitt, 1985b). The unique character of natural evolutional polyploids in acipenseriform taxa makes hybridization between sturgeon species feasible (Birstein, Hanner & De Salle, 1997). Also, the success of interspecific hybridization between sturgeon species is dependent on the ploidy levels of the parent species (Rachek et al., 2022), i.e., hybridization occurs between diploids and diploids, tetraploids and tetraploids, diploids and tetraploids (resulting in triploid hybrids) (Shivaramu, 2019). These hybridizations were mostly reported bidirectional in natural habitats and in domestic farms by artificial fertilization. However, unidirectional hybridization of female Kaluga (Huso dauricus) and Amur sturgeon (Acipenser schrenckii) was reported in natural habitat (Shedko & Shedko, 2016; Shedko, Miroshnichenko & Nemkova, 2020), and viable hybrids were observed only in the hybridization of female Kaluga and male Amur sturgeon. However, in aquaculture conditions viable hybrid offspring were obtained from bidirectional hybridization. Moreover, viable hybrids have been produced from three interspecific crosses through polyspermic fertilization (Iegorova, 2018).

Hybridizations between species from the Family Polyodontidae and species from the Family Acipenseridae were also reported. For example, through artificial fertilization male American paddlefish (Family Polyodontidae) could hybridize with female Russian sturgeon (Acipenser gueldenstaedtii) (Káldy et al., 2020) and with female pallid sturgeon (Scaphirhynchus albus) (Flamio Jr et al., 2021). From the families’ point of view Acipenseridae and Polyodontidae shared the same ancestor, the extinct Family Peipiaosteidae (Birstein & De Salle, 1998), and diverged in the Jurassic period (184.4 Mya) (Peng et al., 2007). In order Acipenseriformes all species have natural polyploids, which contribute to the feasibility of interspecific and intergeneric hybridization such as the hybridization of Family Acipenseridae (Birstein & De Salle, 1998). The polyploid state of acipenserids originates from up to three allopolyploid/autopolyploid whole-genome duplications (WGDs), while the polyodontids originate from one autopolyploid WGD event (Lebeda et al., 2020). The allopolyploid origin contributes to the formation of viable dispermic androgenetic hybrids between phylogenetically close sturgeon species (Grunina & Recoubratsky, 2005). Although, the maternal-effect genes can affect the early embryogenesis before zygotic gene activation (Kishimoto et al., 2004), normal early embryogenesis occurs in closely related sturgeon species with a paternal genome containing homologous silent or not compatible genes (Grunina & Recoubratsky, 2005).

Next to the evolutionary differences, the two species are separated geographically and biologically as well. The sterlet is a Ponto-Caspian distributed species, the smallest sturgeon in the family Acipenseridae, which is a typical bottom feeder (benthic) fish (Fieszl et al., 2011). The American paddlefish is a zooplankton feeding species, native to North America, which is reaching up to 2 m in length and 70 kg in body weight (Mims & Shelton, 2015). Both species are important for aquaculture production and are farmed in ponds and intensive systems in many countries around the world. The eggs of these species are also used to produce black caviar, but their meat is the important market product. Both species are listed on the International Union for Conservation of Nature (IUCN) Red List; the sterlet as “endangered” and the American paddlefish as “vulnerable” conservation status (http://iucnredlist.org/).

The goal of this study was to explore the feasibility of hybridization between two functionally diploid acipenseriform species from different families: the American paddlefish and the sterlet. The objectives were to: (1) examine the fertilization; (2) test the hybrid offspring viability, and (3) analyse of the parentage by use of microsatellite markers in bidirectional hybridization crosses.

Materials & Methods

Ethics

The ethics of these experiments were reviewed and approved by the Békés County Government Office of the Institutional Animal Care and Use Committee of the Research Institute of Aquaculture and Fisheries 179/2020 (license number BE/25/4302-3/2017).

Animals

The parental fish used in the study were obtained from the live ex-situ gene bank of the Research Centre for Fisheries and Aquaculture, Institute for Aquaculture and Environmental Safety, Hungarian University of Agriculture and Life Sciences (Békés County, Szarvas, Hungary). Following the completion of the study, both parental fish and the offspring were placed in closed fish-rearing systems at the same research centre. No euthanasia was applied throughout the course of the experiment.

Experimental design

Two female sterlet (ST1 and ST2) and two male American paddlefish (PF1 and PF2) were crossed to produce four hybrid families ST1 × PF1, ST2 × PF1, ST1 × PF2, and ST2 × PF2. For the reciprocal crosses, one female American paddlefish (PF3) and one male sterlet (ST3) were used to produce one hybrid cross (PF3 × ST3). Within species crosses for sterlet sturgeon (ST1 × ST4, ST2 × ST4) were produced using the two female sterlets (ST1 and ST2) and one male sterlet (ST4), and that for American paddlefish (PF4 × PF1, PF4 × PF2) were produced using one female American paddlefish (PF4) with the two male American paddlefish (PF1 and PF2) (Supplemental Information 1). The sterlet breeders exhibited body weight ranging from 5 to 6 kg and a total body length of 100–120 cm and the American paddlefish showed body weight of 8 to 10 kg and a total body length of 150–200 cm.

Propagation

The propagation processes were executed as previously described in Káldy et al. (2020). To induce ovulation and spermiation, breeders were injected with luteinizing hormone-releasing hormone (LHRH) analog Des-Gly10(D-Ala6) LHRH-ethylamide (Thermo Fisher Scientific, Waltham, USA) at 40 µg/kg for female and 20 µg/kg for male American paddlefish and 10 µg/kg for female and male sterlet. Between 24 and 40 h post injection, milt and eggs were collected by gently stripping the abdomen of breeders at 16 ± 1.0 °C water temperature. Before fertilization, milt was diluted with fresh water at a ratio of 1:200, and the diluted sperm suspensions (2 ml) were immediately mixed with 300-g egg samples according to the common aquaculture practice. Fertilization rates were determined at 70–72 h post-fertilization by counting at least 300 to 350 pieces of eggs. Survival rates were recorded at hatching by counting 50 eggs three times for each cross, and offspring viability was calculated 30 days after hatching.

Culture of fry

600 to 1,000 newly hatched fry from each group were stocked into 400-L tanks with flow-through water for nursing at 18 ± 2.0 °C. In the weaning period, fry was initially fed a combination of Artemia nauplii and chopped Chironomus larvae and at 20 day of post-hatch (DPH) pelleted feed (Aller Infa 0.1–0.4 mm; Aller Aqua Group, Christiansfeld, Denmark) ad libitum. Subsequently, the juvenile fish were exclusively fed with sturgeon feed (Aller Futura 0.5–2.0 mm; Aller Aqua Group) according to their size with ad libitum access at two-hour intervals. The juveniles were housed in four 250-L larval tanks in recirculating larval rearing systems. Once they reached three cm total body length, the juveniles were transferred into four pieces, 1-m3 rearing tanks in a recirculating aquaculture system (RAS); juveniles were fed ad libitum with pelleted fish feed (Aller Performa 1.3–1.5 mm; Aller Aqua Group, Christiansfeld, Denmark) appropriate for their size at 4-hour intervals. When reaching 15–20 cm total body length, fish were stocked into a pond system, and fed with sturgeon feed (Aller Bronze 2.0–4.5 mm; Aller Aqua Group) twice daily.

Microsatellite marker analysis

Sampling and DNA extraction

Five days post fertilization five pooled samples of whole eggs (10 embryos/pool) were collected for microsatellite analysis of viable hybrid embryos and fin clips from the six parents and 12 hybrid individuals (of 15–20 cm in body length) were collected and stored in 96% ethanol at −20 °C until DNA extraction.

For hybrid embryo pools the traditional salting-out method (Miller, Dykes & Polesky, 1988) was applied and modified for fish eggs. Tissue samples were incubated with 300 µl nuclei lysis buffer (1 M Tris-HCl, 0.4 M NaCl, 1 mM EDTA, pH 8.2), 10% SDS and 0.3 mg proteinase K enzyme overnight at 56 °C. Then 100 µl oversaturated NaCl (6 M) was added to each tube followed by centrifugation. DNA was precipitated with 300 µl isopropanol and washed with 500 µl 70% ethanol. After drying, DNA was allowed to dissolve overnight at 37 °C in 60 µl distilled water and stored at 4 °C until use. For DNA isolation from tissue samples E.Z.N.A. Tissue (Omega Bio-Tek, USA) Kit was used according to the producer’s recommendation and stored in ethanol at −20 °C until use.

Microsatellite genotyping

The genotyping was performed with three microsatellite markers (PSP-28, PSP-29, and PSP-32) using a universal fluorescent labeling method (Heist et al., 2002; Zou, Wei & Pan, 2011). A 17-bp long universal sequence tail (5′-ATTACCGCGGCTGCTGG-3′) was linked to the 5′ end of the forward primers, and a tail oligo labeled with different fluorescent dyes (FAM, VIC, NED) (Shimizu et al., 2002) was added to the PCR reactions. The final volume of 20 µL PCR master mix contained 100 ng genomic DNA, 10 X Dream Taq Buffer (Thermo Fisher Scientific, Waltham, MA, USA) with 20 mM MgCl2, 0.2 mM of each dNTPs, 10µM of each primer (forward, reverse, tail) and 1 U Dream Taq DNA polymerase. The PCR profile was: initial denaturation at 94 °C for two min followed by 35 cycles of amplification: 94 °C for 30 s: 56 °C (PSP-28, PSP-29) or 60  °C (PSP-32) annealing for 30 s: 72 °C for 1 min, and with a final extension at 72 °C for 5 min. PCR products were detected using capillary gel electrophoresis system (ABI Prism 3500 Genetic Analyzer, Applied Biosystems) according to the manufacturer’s instructions. GeneMapper 4.0 Analysis software (Applied Biosystems) was used to determine the length of the detected alleles (Káldy et al., 2020).

Chromosome analysis

Chromosome preparation of larvae was made based on a method previously described in Miskolczi et al. (2005). Briefly, about 50 non-feeding larvae were used for chromosome preparations. Approximately fifty larvae were incubated in 0.05% colchicine (Gibco 15212012) for three h, then in distilled water as a hypotonic solution for 25 min. Individual larvae were fixed in a 3:1 ratio of methanol: acetic acid and each larva was homogenized one by one into cell suspension in 50% acetic acid. The suspensions were spread onto pre-cleaned slides and stained in 5% Giemsa (Sigma 48900) solution (in phosphate buffer pH 7.0) for 7–8 min. Chromosome observation was conducted at 1,200× magnification of the well-spread metaphases and numbers were counted.

Statistical analysis

Data were analyzed as previously described in Káldy et al. (2021). Specifically, R Studio Team (version: 2022.7.2.576) and R software (version: 4.3.1 (R Core Team, 2023)) were used to build a logistic regression model (generalized linear model) in which ‘fertilized eggs ratio’ was the response variable, and ‘parent combination’ was the predictor variable. A significant effect of the predictor variable was identified with the ratio of fertilized eggs, and subsequently multiple pairwise comparisons of means (Tukey’s comparison) were performed to further analyze the differences among groups. The following packages were used for the analysis (within the ‘base’ R environment): ‘arm’, ‘ggplot2’, ‘multcomp’. P < 0.05 was considered significant (* P < 0.05, **P < 0.01, ***P < 0.001).

Results

Overall comparison of hybrid and purebred fishes

In comparison, the purebred crosses produced significantly higher fertilization and larval survival than that of all hybrid crosses (Fig. 1). Hybridization crosses yielded variable offspring survivals and the crosses with female paddlefish and male sterlet did not show embryo development to the early neurula stage and thus no percentage of fertilization was collected.

Figure 1 Percentage of fertilized eggs for the parent combinations.

Percentage of fertilization and survival in hybrid crosses of female sterlet (ST) and male paddlefish (PF) and their purebred controls. The crosses were annotated as female individuals (#1 and #2) x male individuals (#1 and #2), such as ST2 x PF1, which means female ST number 2 crossing with male PF number 1. (I) The hybrid; (II) the controls.

On the boxplots, the middle line represents the median value; and the box borders show the lower and upper quartiles. Different letters indicate significant differences between groups regarding fertility (blue) or survival (green). After fitting a logistic regression model, multiple comparisons of means (Tukey’s contrasts) were performed to analyze the differences between groups. P < 0.05 was considered significant (* P < 0.05, **P < 0.01, ***P < 0.001).

Hybrid crosses with different individuals

The percentage of fertilization was significantly different among the parent combinations. The two species of purebred crosses resulted higher value of percentage of fertilization than that of the hybrid crosses (Fig. 1).

Reciprocal hybrid crosses

The reciprocal crosses (male ST3 × female PF3) did not yield viable offspring, although we detected sterlet DNA (ST3) in fertilized American paddlefish eggs (PF3) at 120 h post-fertilization (Table 1). In control purebred American paddlefish cross, the eggs from the PF3 individual fertilized by American paddlefish sperm produced viable purebred American paddlefish offspring (Supplemental Information 1).

Table 1 Analyses of genotypes of hybrid offspring embryos and individuals harvested from the hybridization of sterlet (Acipenser ruthenus; ST) and American paddlefish (Polyodon spathula; PF).

All individuals were confirmed as hybrids.

Microsatellite marker (loci)	PSP-28	PSP-29	PSP-32	
Allele size (bp)	221	242	260	267	178	200	211	225	243	196	242	
ST1♀	221/221					200/200					242/242	
ST2♀	221/221					200/200					242/242	
PF1♂			260	267			211/211	225/225		196/196		
PF2♂			260	267			211/211	225/225		196/196		
HL1	221			267		200	211	225		196	242	
HL2	221		260			200	211	225		196	242	
HL3	221			267		200	211	225		196	242	
HL4	221			267		200	211	225		196	242	
HL5	221		260			200	211	225		196	242	
HL6	221			267		200	211	225		196	242	
HL7	221			267		200	211	225		196	242	
HL8	221			267		200	211	225		196	242	
HL9	221			267		200	211	225		196	242	
HL10	221			267		200	211	225		196	242	
HL11	221			267		200	211	225		196	242	
HL12	221			267		200	211	225		196	242	
ST3♂	221/221					200/200					242/242	
PF3♀		242	260		178		211	225	243	196/196		
HE1	221	242				200	211	225		196	242	
HE2	221	242	260			200	211	225	243	196	242	
HE3	221		260		178	200	211	225	243	196	242	
HE4	221	242				200	211	225		196	242	
HE5	221		260			200	211	225		196	242	
Notes.

ST1 and ST2 are sterlet females, ST3 is the sterlet male used; PF1 and PF2 are American paddlefish males, PF3 is the American paddlefish female used; HL1-12: samples from hatched hybrid larvae; Pool-HE1-5: pooled samples from hybrid embryos (10 embryos/pool); sterlet alleles are marked in italics.

Chromosome observation in hybrid larvae

The karyotypic analysis of the viable hybrid individuals showed tetraploid karyotypes (chromosome number ∼120), similar to the parent species.

Genotypic analysis of hybrid offspring

Genotypes of putative hybrids by three microsatellite markers (Káldy et al., 2020) indicated that none of the used markers shared the same alleles between the two parent species from the hybrid crosses. However, the genotypes of the two males (PF1 and PF2) were identical, as well as for the two females (ST1 and ST2). All hatched hybrid larvae (n = 3,228) exhibited one or two alleles inherited from their parents. In hybrid crosses of American paddlefish female × sterlet male, the genotype of pooled embryos showed one to four alleles originated from the parents. Altogether, eleven alleles were detected by the three markers in the functional tetraploid parental group, which consisted of 6 individuals. On the PSP-29 locus had five alleles, followed by PSP-28 with four alleles, and the PSP-32 locus had the fewest, two alleles. In the larval hybrids, eight alleles were detected, while in the hybrid embryos pools ten alleles. Considering the tetrasomic allele distribution observed at the PSP-29 marker in American paddlefish, the segregation patterns aligned with our expectations. The results showed that the hybrid larvae as well as the hybrid embryo pools inherited alleles from both parents, and successful fertilization was confirmed in both directions of crosses (Table 1).

Morphologically, hybrid offspring exhibited characteristics of both parents, such as a lack of the scutes that are typical in sturgeon species. In terms of the body shape, the hybrid offspring were similar to sturgeon species but not paddlefishes.

Discussion

Hybridization among paddlefish and sturgeons has been attempted before. For example, Mims et al. (1997) reported that American paddlefish eggs could be fertilized with shovelnose sturgeon (Scaphirhynchus platorynchus) sperm and that the fertilized eggs developed to the gastrula stage. While, Zou, Wei & Pan (2011) showed that embryogenesis reached the gastrula and early neurula stages for American paddlefish eggs fertilized by Amur sturgeon (Acipenser schrenckii) sperm. Overall hybrid crosses with female paddlefish and male Acipenseridae sturgeon did not result in viable offspring, but in contrast, more reciprocal hybrid crosses between these taxa, resulted in viable hybrid offspring. There are several successful American paddlefish gynogenesis studies used by inactive Amur sturgeon (Zou, Wei & Pan, 2011) and shovelnose sturgeon (Mims et al., 1997) sperm. However, gynogenesis could be successful not only with inactivated sperm, but also with active foreign sperm as the natural gynogenetically reproductive form of silver crucian carp (Carassius auratus) (Fan & Liu, 1990). After (normal or gynogenetic) fertilization, the male and female pronuclei fuse to form the zygotic nucleus; the second meiosis resumes first and then the zygote enters the first mitotic division stage (Lindemanand & Pelegri, 2010). If the invasive sperm caused gynogenesis, the paternal genome in the fertilized egg is lost before the first mitotic division (Beukeboom & Vrijenhoek, 1998). The first mitotic division in the paddlefish egg occurs about 180 min after fertilization at 16 °C (Mims & Shelton, 2015), and gastrulation begins approximately after day two, and neurulation lasts until day six (Bemis & Grande, 1992). Our results show that in American paddlefish eggs fertilized by sterlet sperm, the paternal chromatin was present in early cell cleavage and also in the early embryonic stage (not lost before first mitotic division), because the paternal genome was detected 120 h post fertilization. It suggests that unidirectional hybridization is not caused by the death of the unfertilized haploid egg (for example in gynogenesis by hybridization), but by a genetic process or processes that occur during embryogenesis.

In embryogenesis, gastrulation is the first major block of development (Merriner, 1971), where the eggs without paternal nuclei die during gastrulation. The genetic incompatibilities between gametes is less investigated in fishes than in other animal species, but it has been established that this incompatibility enables hybridization in one direction while hindering it in the other direction, thereby resulting in unidirectional hybridization (Álvarez & Garcia-Vazquez, 2011).

Within this, the genetic explanation may be an abnormal gene expression (regulation). If the mutational differences between species reach a saturation point, gene regulation/expression divergence between species will be too large for the embryos to survive the gastrula stage (Parker, Philipp & Whitt, 1985a). For example, hybridization between species belonging to the genus Lepomis (i.e., green sunfish Lepomis cyanellus) and Micropterus (i.e., largemouth bass Micropterus salmoides) (Centrarchidae, Perciformes, Teleostei) led to unidirectional hybridization, which was described as being caused by normal gene expression in one direction and an abnormal pattern of gene regulation in the other direction, leading to embryo lethality (Whitt, Philipp & Childers, 1977). This result highlights that the gene expression of related species does not necessarily have the same molecular and genetic basis, so when genes from different species are placed in a common nucleus, their mutational differences can lead to incompatibility in the hybrid offspring (Whitt, Philipp & Childers, 1977), or affect the transcription of multiple genes, which may induce stress signalling (Landry, Hartl & Ranz, 2007). Altered gene expression may also be caused by repetitive DNA and inappropriate activity of silenced genes, which may be caused by defects in several epigenetic mechanisms in hybrids (Maheshwari & Barbash, 2011). The gene expression in the early development of embryos is regulated by the maternal cytoplasm; in the case of hybridization of triangular bream (Megalobrama terminalis) and grass carp (Ctenopharyngodon idellus), the hybrid is viable if the maternal species is grass carp, otherwise, the hybrid offspring will not develop normally (Liu et al., 2020). This is not due to the incompatibility of the nucleic DNA of the parent species, but to the abnormal coordination of the nuclear DNA of the triangular bream and the sperm of the grass carp (Liu et al., 2020). In addition, the divergence of gene expression between increasingly distant species increases further (Philipp, Childers & Whitt, 1979; Parker, Philipp & Whitt, 1985b).

Further genetic explanation could be the cytogenetic maternal influence, namely by the transcription of maternal mRNA into functional proteins in the embryo, which can affect the development of the embryo in the early stages of embryogenesis. Embryonic development begins under maternal regulation of mRNA and protein synthesis, and then this regulation shifts to zygotic transcript levels (referred to as the maternal-zygotic transition). However, incompatibilities can then arise between paternal alleles and the maternal factors that halt embryonic development (Turelli & Moyle, 2007). The paternal genome does not have genes that are appropriately silenced to allow full expression of maternally derived mRNA, and poor interaction of maternal and paternal genes can impede development (Grunina & Recoubratsky, 2005). But genetic imbalances may be caused by uniparental mitochondria, mRNAs, proteins and noncoding RNAs inherited from maternal cytoplasm, which may affect epigenetic processes that critically depend on maternally inherited factors (Maheshwari & Barbash, 2011), such as when maternally inherited material fails to silence embryonic lethality induced by paternally inherited blockade of satellite DNA (Maheshwari & Barbash, 2011).

On the other hand, crossing a male American paddlefish with a female sterlet resulted in viable hybrid offspring, as did crossing a male American paddlefish with a female Russian (Káldy et al., 2020) or pallid (Flamio Jr et al., 2021) sturgeon. The chromosome number of the hybrids was identical to the chromosome number of the parent species, as in the case of hybrid offspring with pallid sturgeon, where the chromosome number of the parent species was also identical (Flamio Jr et al., 2021). In contrast, the hatching rate of viable hybrids of sterlet and paddlefish is much lower than that of purebred sterlet and paddlefish. This is not surprising, as there is already a big difference in successful crossing between inter- and intrageneric hybridization. The hatching rates between salmonid species were equivalent to the purebred control in 18 out of 43 intrageneric hybridizations, while in intergeneric hybridization, it was equivalent to the purebred control in three out of 46 hybridizations (Chevassus, 1979).

Conclusions

The hybridization between female sterlet and male American paddlefish was proved with viable verified offspring harvested, but embryos from reciprocal hybridization do not survive the stage of gastrula or early neurula. To date, successful hybridization with viable offspring between the Acipenseridae and Polyodontidae families has been reported only when the American paddlefish were applied as paternal parents. Possibly, the unidirectional hybridization observed in this study was attributed to genetic incompatibility during early embryonic development (and not to the fertile incompatibility), as indicated by microsatellite marker analysis. Further investigation of the unidirectional hybridization is warranted.

Supplemental Information

Supplemental Information 1 Experimental design and embryo development with percentage of fertilization and survival of hybrids (sterlet Acipenser ruthenus× American paddlefish Polyodon spathula) and purebred families

Percentage of fertilization = number of live eggs at neurulation/100 eggs; percentage of survival = number of live fish at 30 days post hatch/100 eggs.

Click here for additional data file.

Supplemental Information 2 Raw details

Click here for additional data file.

We are thankful for the help and support of Béla Halasi-Kovács and Miklós Bercsényi during the work required for hybridization of fish and data collection.

Additional Information and Declarations

Competing Interests

Author Contributions

Animal Ethics

Data Availability

The authors declare there are no competing interests.

Jenő Káldy conceived and designed the experiments, prepared figures and/or tables, authored or reviewed drafts of the article, and approved the final draft.

Georgina Fazekas performed the experiments, prepared figures and/or tables, and approved the final draft.

Balázs Kovács performed the experiments, authored or reviewed drafts of the article, and approved the final draft.

Mariann Molnár analyzed the data, prepared figures and/or tables, and approved the final draft.

Bence Lázár analyzed the data, prepared figures and/or tables, and approved the final draft.

Nóra Pálinkás-Bodzsár analyzed the data, prepared figures and/or tables, and approved the final draft.

Uroš Ljubobratović performed the experiments, prepared figures and/or tables, and approved the final draft.

Gyöngyvér Fazekas analyzed the data, prepared figures and/or tables, and approved the final draft.

Gyula Kovács performed the experiments, authored or reviewed drafts of the article, and approved the final draft.

Eszter Várkonyi conceived and designed the experiments, performed the experiments, authored or reviewed drafts of the article, and approved the final draft.

The following information was supplied relating to ethical approvals (i.e., approving body and any reference numbers):

Experiments were ethically approved and reviewed by the Institutional Animal Care and Use Committee of the Research Institute of Aquaculture and Fisheries 17-9/2020 (license number BE/25/4302-3/2017).

The following information was supplied regarding data availability:

The raw data is available in the Supplemental Files.

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
