# Peer review of "Unidirectional hybridization between American paddlefish Polyodon spathula (Walbaum, 1792) and sterlet Acipenser ruthenus (Linnaeus, 1758)"

_PeerJ, doi:10.7717/peerj.16717_

## Round 0.1 · original submission · Major Revisions

The manuscript should be thoroughly revised following all suggestions and recommendations made by the reviewers. Overall, there are 3 main concerns:

- English needs to be revised and improved

- Some more details need to be added to the materials and methods to increase the reproducibility for the reader.

- The conclusion should be rewritten in order to describe better the obtained results.

**Language Note:** The Academic Editor has identified that the English language must be improved. PeerJ can provide language editing services - please contact us at copyediting@peerj.com for pricing (be sure to provide your manuscript number and title). Alternatively, you should make your own arrangements to improve the language quality and provide details in your response letter. – PeerJ Staff

Reviewer 1 ·

Basic reporting

It is amazing, but many fishes can produce viable interspecific hybrids, a few can produce intergeneric hybrids, and a very few even interfamilial hybrids. Kaldy et al. produced hybrid paddlefish x starlet, noting that one cross was viable and the reciprocal cross was inviable. They showed inheritance of microsatellites to show that both genomes were present in the young, and counted chromosomes to determine ploidy. This is all well and good, though the text can use work to bring the scientific prose up to the expected standard; I have marked the manuscript to assist with that. The Discussion needs considerable work to better explain the results and to place them in the broader context; more on that below.
Abstract. – At line 30, “other” must be deleted. Paddlefish is not a sturgeon.
At line 32, we are not speaking of “backcross” hybrids here – a backcross is the result of an F1 or later-generation hybrid being crossed with one of the original species. Here the authors are writing of a “reciprocal” hybrid. The terminology should be crisp.
Introduction. – At line 47 is an incomplete sentence: The diploid chromosome count of the common ancestor of Acipenseriformes was approximately… what? 60?
At line 49, the authors write of three karyotypes, which can be clarified. Does this mean three types of ancestry, allotetraploid, autotetraploidy and ??, or just what?
The sentence now appearing at lines 62-64 would serve well as the topic sentence for the paragraph starting at line 56. The sentence can be better supported by adding other citations, perhaps Parker et al. (1985a, b) and more recent works that cite Parker et al. (1985) and Arnold et al. (1997). This is a key passage, setting up the Discussion later, so investment here would be worthwhile.
Methods. – At line 105, what is meant by “hooks”? Could there be a poor translation here?
At line 118, these are not backcross, but rather reciprocal crosses.
At line 119, I’d urge the authors to write of five hybrid families; to write of “combinations” suggests different interspecific crosses. The authors should mention that they made (at least) one within-species paddlefish cross, which is important to show near-complete fertilization success within that species. Do they also have data for within-species crosses within starlet? That would be useful for claiming near-complete fertilization success within that species. Otherwise, the comparisons within Figure 1 lack comparators and are less convincing.
The microsatellite genotyping section is just too short. The reader should be told the names of the three microsatellite loci, which will be useful for understanding Table 1.
At line 161, what analytic package within the R environment was used? To say it was applied within R is useless to those that would want to build upon this work.
Results. – The first paragraph of the Results section should tell the reader that the observed fertilization percentages for the hybrids were much lower than those for within-species crosses. The comparison should be made within Figure 1 as well.
Discussion. – The section should open with an appropriate topic sentence. Perhaps: Other workers have attempted hybridization among paddlefish and sturgeons.
At line 213, use of the word “abortion” is in appropriate; it would be better to write of the failure of embryos to develop.
One of my major critiques of this manuscript is that its scope is so narrowly focused on acipenseriform fishes that it ignores reports of studies of other fish lineages whose results may give perspective on these results and perhaps also explain them. Chevassus (1979) reported the results of crosses between species and general of salmonids and Parker et al. (1985a, b) draw on extensive work by Childers and Whitt to assess factors affecting developmental success in hybrid centrarchids. At line 217, discussion of these papers and more recent works citing them – hopefully drawing in molecular work on developmental pathways – would greatly strengthen the Discussion and the paper more generally.
The last paragraph of the Discussion gets at molecular mechanisms underlying altered development of hybrid embryos. It is hard to follow and I’m not sure that it reflects current thinking adequately. Let’s consider this passage a bit at a time.
Line 218 can be better cast as: The maternal effect, which is explained by the transcription of maternal mRNA into functional proteins in the embryo, can influence its development…
The sentence about the Grunina and Recoubratsky explanation could be more crisply rendered, saying that the paternal genome does not have genes that are appropriately silenced to allow full expression of maternally derived mRNA, and that poor interaction of maternal and paternal genes can impede development.
At line 223, what is meant by “ancient state”? Of paternal gene expression?
Why are the paragraphs at lines 218 and 223 separated? They both seem to seek to explain the differential outcomes of the two hybrid crosses.
This whole paragraph should be redrafted after consulting recent literature, such as Parker et al. (1985a, b), Landry et al. (2007), Turelli et al. (2007), and papers citing these works.
References. – I ‘ve marked minor stylistic issues in the literature citations.
Tables and Figures. – In Figure 1, those are not “ratios” (2:1, 3:1, etc.), but rather percentages. The percent fertilization of within-species hybrids should also be presented to provide comparators for the results shown.
In Table 1, the reader should be told that Psp 28, 29, and 32 are microsatellite loci.
Literature cited in this review:
Chevassus B. 1979. Hybridization in salmonids: results and perspectives. Aquaculture 17(2):113-128.
Landry CR, Hartl DL, Ranz JM. 2007. Genome clashes in hybrids: insights from gene expression. Heredity 99(5):483-493.
Parker HR, Philipp DP, Whitt GS. 1985a. Gene regulatory divergence among species estimated by altered developmental patterns in interspecific hybrids. Molecular Biology and Evolution 2(3):217-250.
Parker HR, Philipp DP, Whitt GS. 1985b. Relative developmental success of interspecific Lepomis hybrids as an estimate of gene regulatory divergence between species. Journal of Experimental Zoology 233(3):451-466.
Turelli M, Moyle LC. 2007. Asymmetric postmating isolation: Darwin's corollary to Haldane's rule. Genetics 176(2):1059-1088.

Experimental design

The experimental design is basically fine, although it would be strengthened with additional of data for fertilization success for the respective within-species crosses.

Validity of the findings

The scope of the experiment is narrow, and the results are valid within that scope. Greater contextualization would strengthen the manuscript greatly.

Annotated reviews are not available for download in order to protect the identity of reviewers who chose to remain anonymous.

Reviewer 2 ·

Basic reporting

This manuscript reported the hybridization crosses of American paddlefish Polyodon spathula and starlet sturgeon Acipenser ruthenus in comparison to the pure crosses. I believe that the authors have done a lot of work on the broodstock culture and spawn, gamete collection and fertilization, and larval and juvenile culture. This manuscript needs to be expanded with more detailed information. Furthermore, many statements in the manuscript need to be revised for clarification (see the detailed comments in the PDF file).

Experimental design

The experimental design is sound to support the research goal.

1) The fertilization data in the supplementary file should be organized and reported in the manuscript in Figure 1 (or use a Table to hold more information). For each cross and each breeding type, the fertilization varied and needs to be reported in detail. In fact, the marker analysis in Table 2 did this kind of report. Additionally, the survived juvenile data should be also reported.

Table 1. The fertilization, embryo development, and survival of hybridization between ……. in comparison to the control crosses of paddle …. And …….

2) A Figure or table for the breeding schedule should be added for readers to understand immediately how many crosses the authors produced and what are the parents.

Validity of the findings

Three markers were used in this study for hybridization confirmation. However, further investigation may be required to confirm the Genetic Incompatibility. Therefore, the manuscript title was suggested to be revised.

Additional comments

The detailed review comments on this manuscript were put in the attached PDF file by use of the “PDF comments” tracking system.

Annotated reviews are not available for download in order to protect the identity of reviewers who chose to remain anonymous.

Reviewer 3 ·

Basic reporting

BROAD FEEDBACK ON BASIC REPORTING:
- The introduction requires a restructure to better place the study in the context of broader literature and provide a clear question to be addressed or objective of the article. The introduction draws on relevant literature (although more recent articles could also be appropriate to cite) but is not structure in a logical way. Restructuring this section, starting broad and with clear objectives, will improve clarity and conform to standard formatting conventions.
- The literature cited by authors does not include a number of relevant prior publications on a well studied taxonomic group, the Acipenseridae. Only four of the publications cited are from the past five years. I suggest a broader literature review of hybridization within Acipenseridae to better place the work within the broader field of knowledge.
- The conclusion would have been more immediately impactful if it directly related to a question or knowledge gap outlined in the introduction.

SPECIFIC FEEDBACK:
- Line 40-41: A small suggestion to consider updating keywords to include those words not already found within your title to enhance the impact/reach of your study. At present the majority of key words already feature in the title.
- Line 46-47: sentence incomplete.
- Line 65-68: sentence structure is technically not correct or clear. Try to reword this statement in a shorter and unambiguous way.
- Line 96: consider italicising genera name Artemia
- Line 96: did the author mean 'cut' rather than 'cutted'?
- Line 101: convention is to use whole words for numbers under 10. Change '4' to 'four'. Apply where relevant throughout manuscript.
- Line 117-119: review sentence structure for clarity.
- Line 141: review sentence structure for clarity.
- Line 210: sentence incomplete; review structure for clarity.
- Line 213: review wording.

Experimental design

- Research question not well defined in introduction, so it is not clearly stated for the reader to understand where or how the research addresses a particular knowledge gap.
- Methods described were detailed and included sufficient information to replicate. The authors followed well established and appropriate methodologies and rationale.
- Ethical standards appear to be considered, reviewed and upheld.

Validity of the findings

- Not all underlying data were provided. No sequences or indication of where that data may be made available in an acceptable repository online are indicated.
- Conclusions are somewhat described but lack impact without being connected to an original, clear research question being investigated.
- The conclusion statement was generally vague and missed the opportunity to shed light on the importance of the studies findings.

Additional comments

I commend the authors for providing clear and replicable methods. There is an important question and finding in this manuscript but it is not clearly communicated to the reader. The main way this manuscript could be improved is in refining the writing structure to provide a clear and unambiguous objective and define the knowledge gap to be addressed, firstly through citing more recent relevant literature on the well-studied topic of hybridization in Acipenseridae, and then clearly linking any conclusions drawn from the findings of the study to the objective, highlighting the importance of the research. I think it is a meaningful study with important data however requires significant revision before publication within the literature.

---

## Round 0.2 · Major Revisions

The manuscript needs a detailed and thorough revision because it still has a lot of things to improve and correct. The reviewers' comments are attached in the attached files. Please take all the suggestions into account when drafting the modified version of the manuscript.

Reviewer 1 ·

Basic reporting

Crosses of many fishes can produce viable interspecific hybrids, a few can produce intergeneric hybrids, and a very few even interfamilial hybrids. Kaldy et al. produced hybrid paddlefish x starlet, noting that one cross was viable and the reciprocal cross was inviable. They showed inheritance of microsatellites to show that both genomes were present in the young, and counted chromosomes to determine ploidy. This is a revised manuscript, and is clearly better than the first version. This is especially due to a much-strengthened Discussion. The prose is also improved, though further work will bring the scientific prose up to standard; I have marked the manuscript to assist with that.
Abstract. – At line 37, the Latin name of sterlet should be given.
References. – I ‘ve marked minor stylistic issues in the literature citations, mostly pertaining to the need to Italicize species’ Latin names.

Experimental design

Methods. – I have marked redundant sentences at lines 136-137 and 140, which should be deleted.
A sentence appearing at lines 152-153 should be moved to line 155 to present methods in the temporal order in which they were executed. Similarly, a sentence at lines 159-160 should be moved to line 158. I have marked the manuscript.

Validity of the findings

Discussion. – At line 298, I think that the key issue is not the cytoplasm in which the respective coadapted gene complexes that drive embryogenesis find themselves, but rather their having to function together in the nucleus. The Whitt et al. (1977) explanation seems way out of line with current thinking on this matter.
I also wonder about the defensibility of the Maheshwari et al. (2011) explanation offered at lines 321-323.
At line 332, the authors should include the adjective “salmonid” to provide critical context for the Chevassus citation appropriately used there.

Annotated reviews are not available for download in order to protect the identity of reviewers who chose to remain anonymous.

Reviewer 2 ·

Basic reporting

See the attached files for details.

Experimental design

See the attached files for details.

Validity of the findings

See the attached files for details.

Additional comments

See the attached files for details.

Annotated reviews are not available for download in order to protect the identity of reviewers who chose to remain anonymous.

---

## Round 0.3 · accepted · Accept

Thank you very much for sending the revised manuscript. After reading both the comments made by the reviewers and the rebuttal letter of those changes, I have been able to verify that all the changes suggested by the reviewers have been implemented in the manuscript.

I believe that the current version of the paper is much better than the initial one thanks to the improvements introduced. Therefore, I am pleased to accept the manuscript for publication in PeerJ.

Thank you for submitting your work to this journal.

With kind regards,